# Intrinsic exchange biased anomalous Hall effect in an uncompensated antiferromagnet MnBi$_2$Te$_4$

Su Kong Chong [1,8] ✉, Yang Cheng[1,8], Huiyuan Man[2,3], Seng Huat Lee [4,5], Yu Wang[4,5], Bingqian Dai [1], Masaki Tanabe[1], Ting-Hsun Yang[1], Zhiqiang Mao [4,5], Kathryn A. Moler[2,6,7] & Kang L. Wang [1] ✉

Achieving spin-pinning at the interface of hetero-bilayer ferromagnet/ antiferromagnet structures in conventional exchange bias systems can be challenging due to difficulties in interface control and the weakening of spin-pinning caused by poor interface quality. In this work, we propose an alternative approach to stabilize the exchange interaction at the interface of an uncompensated antiferromagnet by utilizing a gradient of interlayer exchange coupling. We demonstrate this exchange interaction through a designed field training protocol in the odd-layer topological antiferromagnet MnBi$_2$Te$_4$. Our results reveal a remarkable field-trained exchange bias of up to ~ 400 mT, which exhibits high repeatability and can be easily reset by a large training field. Notably, this field-trained exchange bias effect persists even with zero-field initialization, presenting a stark contrast to the traditional field-cooled exchange bias. The highly tunable exchange bias observed in this single anti-ferromagnet compound, without the need for an additional magnetic layer, provides valuable insight into the exchange interaction mechanism. These findings pave the way for the systematic design of topological anti-ferromagnetic spintronics.

Two-dimensional antiferromagnetic (AFM) materials have shown promise for spintronic applications[1–6] due to their ultrafast switching dynamics and the absence of stray fields, setting them apart from ferromagnets. MnBi$_2$Te$_4$ (MBT) is a notable example of A-type AFM with Neel temperature (T$_N$) of ~24 K, and it has garnered significant research interest due to the intriguing interplay between its topology and magnetic orders[7–9]. In MBT, the magnetization primarily originates from the Mn layer, which is inherently integrated into the crystal lattice. The combination of intralayer ferromagnetic (FM) and interlayer AFM coupling gives rise to a unique layer-dependent magnetism in thin film MBT, where the odd and even layers exhibit distinct magnetic and topological properties[10–15]. Specifically, the anomalous Hall effect resulting from the uncompensated layer of the odd-layer MBT can display complex magnetic interactions coupled with its surface band topology.

The magnetic exchange interaction plays a crucial role in the manipulation of the magnetic functionality in magnetic memory and spintronic devices[16,17]. One notable manifestation of the exchange interaction is the exchange bias effect, which can be observed as a horizontal shift in the magnetic hysteresis loop when a magnetic

[1]Department of Electrical and Computer Engineering, University of California, Los Angeles, CA 90095, USA. [2]Geballe Laboratory for Advanced Materials, Stanford University, Stanford, CA 94305, USA. [3]Stanford Nano Shared Facilities, Stanford University, Stanford, CA 94305, USA. [4]2D Crystal Consortium, Materials Research Institute, The Pennsylvania State University, University Park, PA 16802, USA. [5]Department of Physics, The Pennsylvania State University, University Park, PA 16802, USA. [6]Stanford Institute for Materials and Energy Sciences, SLAC National Accelerator Laboratory, Menlo Park, CA 94025, USA. [7]Department of Physics and Applied Physics, Stanford University, Stanford, CA 94305, USA. [8]These authors contributed equally: Su Kong Chong, Yang Cheng. ✉e-mail: sukongc@g.ucla.edu; wang@seas.ucla.edu

material is cooled in the presence of a magnetic field. This effect typically occurs in FM/AFM hetero-bilayers, where the magnitude of the shift is determined by the strength of the pinning exchange interaction at the interface between the FM and AFM layers[18–24]. In addition to FM/AFM hetero-bilayers, the exchange bias effect has also been observed in other types of heterostructures, such as FM/spin glass[25,26] and FM/paramagnetic superlattices[27,28], indicating the complex nature of the interface pinning exchange interaction. Recently, the exchange bias effect has been found to stabilize in single compounds with complex magnetic phases[26,29–31]. The exchange bias effect in single-component magnetic material on a non-magnetic substrate or without an additional antiferromagnetic layer can be a candidate for the intrinsic exchange bias effect[32,33]. An interesting example is bulk $Co_3Sn_2S_2$, which exhibits an exchange bias anomalous Hall effect driven by fluctuations originating from its spin glass phase[29]. However, this exchange bias effect has been found to diminish in thin film $Co_3Sn_2S_2$[34], thereby limiting its investigation in the two-dimensional limit.

In our study, we focus on investigating the exchange interactions in an uncompensated topological AFM MBT thin film. To ensure consistency, we specifically study the 7 septuple-layer (SL) thickness,

as it is the most accessible for our experiments (S.I. Fig. S1 and Table S1). The quality of these devices is confirmed by their large anomalous Hall in the spin-alignment phase in a magnetic field (S.I. Fig. S2). Additionally, we use the magnetic transition fields as an additional verification of layer number[12,15] (S.I. Fig. S3). Due to the weak van der Waals interlayer bonding in MBT, the A-type AFM coupling with an uncompensated FM layer can, in principle, form a quasi-FM/AFM bilayer (Fig. 1a). Depending on the strengths of the interlayer coupling, it can naturally lead to an exchange interaction without the need for an additional AFM layer. Here, we employ electrical transport measurements to investigate the exchange bias effect as they provide an effective means of probing for the exchange interaction. Specifically, Hall measurements give an overall charge response to the magnetic signal, making them compatible with electronic devices for spintronic applications[16]. We discuss two mechanisms for the exchange bias anomalous Hall effect: field cooling and field training methods. Field cooling refers to the cooling cycle performed under a constant magnetic field; while field training involves manipulating the magnetic field without changing the temperature or undergoing a different cooling cycle. We show that the gradient of exchange coupling strength and the domain structure

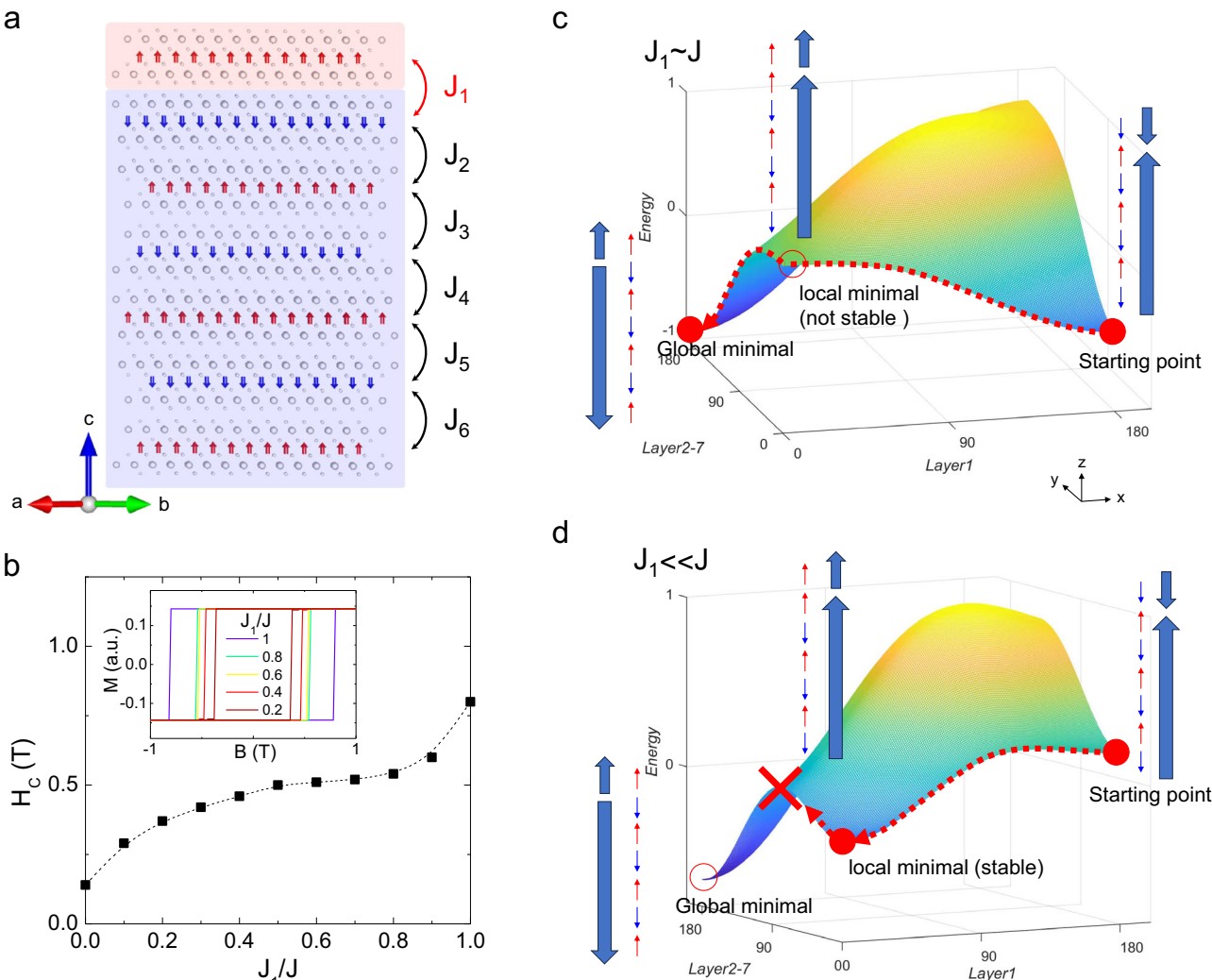

**Fig. 1 | Manipulation of the exchange coupling in an uncompensated anti-ferromagnet. a** Schematic of the atomic structure for a 7SL MBT AFM with the nearest neighbor exchange coupling is denoted in the figure. **b** Micromagnetic simulation for the coercive field, $H_C$, as a function of the exchange coupling ratio, $J_1/J$, for the MBT. The inset in b is the simulated magnetization versus magnetic field

at different exchange coupling ratios. Schematic of energy landscapes and trajectory of the spin states in 7SL MBT when evolving from negative to positive field for **c** comparable exchange coupling ($J_1 \sim J$), and **d** weak coupling of the first layer to the total exchange coupling ($J_1 \ll J$). The x-axis represents the spin angles in layer 1, while the y-axis represents the spin angles in layers 2–7 of the 7SL MBT.

of the uncompensated FM layer play crucial roles in the exchange bias effect.

## Results and discussion
### Quasi-FM/AFM bilayer
Figure 1a shows the schematic of the atomic structure of a 7SL MBT with the $J_i$ representing the magnetic exchange coupling between adjacent layers as labeled in the figure. In an ideal A-type antiferromagnet with the same nearest-neighbor exchange interaction, the spin Hamiltonian can be expressed as $H = -J \sum_{ij} S_i \cdot S_j - K \sum_i (s_i^z)^2$, where $J$ and $K$ describe the interlayer exchange interaction between spins at sites $i$ and $j$, and the intralayer magnetic anisotropy of spins at site $i$, respectively[13,35]. However, due to the imperfections in crystal growth and/or degradation in the device preparation process, the MBT thin flakes can contain structure defects and domains, which can weaken the exchange coupling strength of the outermost uncompensated layer ($i = 1$). In Fig. 1b, micromagnetic simulation of a 7SL MBT is shown by varying the ratio of the first layer to total exchange coupling ($J_1/J$). The simulated magnetization versus magnetic field curves exhibit a systematic reduction in the coercivity with the decreasing $J_1/J$. As the interlayer exchange energy is proportional to $J_1 S_1 S_i$, a decrease in $J_1$ leads to a reduction in the coercive field. Figure 1c, d illustrates the energy of spin states by setting the interlayer exchange coupling of the uncompensated layer to $J_1 \sim J$ and $J_1 \ll J$, respectively. For $J_1 \sim J$, the only stable spin state is the anti-parallel coupling state, as indicated by the filled red circles in Fig. 1c. However, in the case of $J_1 \ll J$, the energy of the parallel coupling state decreases to a local minimum, forming a metastable state with the parallel spin state of the first two layers. Under such circumstances, unless a large amount of energy is supplied, the transition to the global minimal ground state is prohibited. Therefore, the odd SL MBT can be expected to behave as a quasi-FM/ AFM bilayer.

## Magnetic domains in the uncompensated layer
To investigate the magnetic structure of the MBT, we employed a scanning superconducting quantum interference device microscope (SSM). Figure 2a presents the optical image and SSM mappings of a multilayer MBT flake exfoliated onto a Si/SiO₂ substrate. The magnetization signals were only detected from the odd layer region of MBT rather than the even layer region. This confirms the magnetization signal observed in our SSM originated from the uncompensated FM layer in the odd SL MBT, which is consistent with other A-type AFM with an out-of-plane direction as an easy axis, such as CrI₃[36]. To study the magnetic domains in the odd SL MBT, we performed measurements under multiple field cooling cycles, as described in Fig. 2a. Figure 2a(ii)−(vi) demonstrates the development of multidomain states during zero-field cooling. Under different zero-field cooling cycles, the arrangement of up and down domains appeared random. However, a single domain state was achieved for all odd layer thicknesses by field cooling at a very small magnetic field of 3 (−3) mT applied along the easy (out of plane) axis. It is worth noting that this field is much smaller than the coercive field of >100 mT at 4 K. This indicates that the magnetic spin can be magnetized at a very small magnetic field near $T_N$.

## Field cooling exchange bias
Next, we study the magnetic properties of the MBT devices using a standard Hall measurement. Figure 2b shows the optical image of a typical 7SL MBT device fabricated in a Hall bar configuration. The AFM coupling in these devices is confirmed by the magnetic transitions observed at high magnetic fields (S.I. Fig. S2). Additionally, the uncompensated layer in MBT primarily contributes to the $R_{yx}$ hysteresis loop around zero magnetic field. It should be noted that the $R_{yx}$ hysteresis loops were obtained with the gate voltage controlled to slightly below the charge neutrality point. This was done to prevent any mixing of the longitudinal resistance caused by resistive contacts

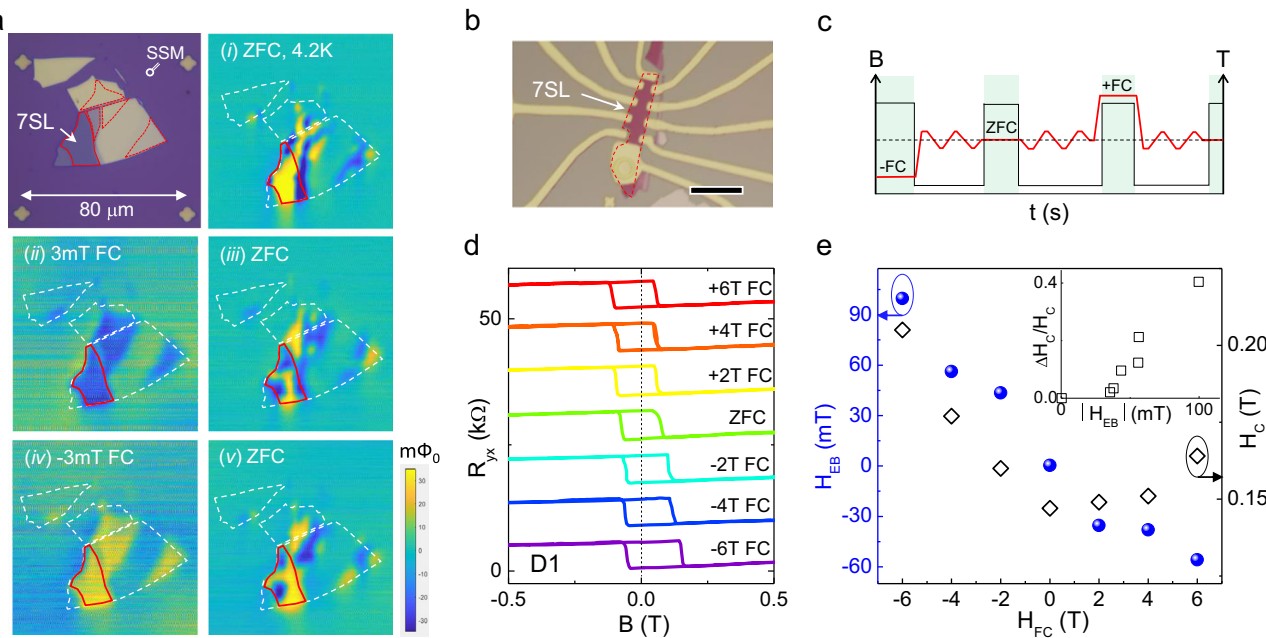

**Fig. 2 | Magnetic domains and field cooling exchange bias. a** Optical image and low-temperature SSM mappings for variable thicknesses MBT flake. The SSM measurement was performed following the sequence at a base temperature of 4.2 K: (i) ZFC, (ii) 3 mT FC, (iii) ZFC, (iv) −3 mT FC, and (v) ZFC. The white dashed lines trace the region of MBT flakes. The red lines trace the region of 7SL MBT. Multi-domains form at ZFC with the up and down domains distinguished by blue and yellow colors, respectively. **b** Optical image of a representative 7SL MBT device fabricated in Hall bar geometry. The scale bar is 20 μm. **c** Illustration of the field cooling protocol for the setting of temperature (black) and magnetic field (red) as a function of time. The green shades highlight the regions for field setting and cooling. **d** $R_{yx}$ hysteresis loops in magnetic field measured at different field cooling cycles at 2 K for device D1. Curves in **c** are vertically shifted for comparison. **e** Plots of exchange bias field ($H_{EB}$) and coercive field ($H_C$) as a function of cooling fields ($H_{FC}$) for device D1. Inset in **e** plots the $\Delta H_C/H_C(0)$ versus $|H_{EB}|$ for device D1.

at the charge neutrality point (S.I. Fig. S2). Therefore, the positive slope observed in the saturation region of the $R_{yx}$ versus magnetic field curves (S.I. Fig. S3) can be attributed to the ordinary Hall effect arising from the hole carriers.

We examine the field cooling effect on the MBT device D1. The temperature and magnetic field settings for the field cooling sequence are illustrated in Fig. 2c. Figure 2d displays the $R_{yx}$ hysteresis loop measured at different field cooling cycles. When cooled at different magnetic fields ranging systematically from −6 T to +6 T, the hysteresis loop gradually shifts from positive field extension to negative field extension, indicating a negative exchange bias effect. The shift in the $R_{yx}$ hysteresis loop is a characteristic of the exchange bias effect, which shares similarities with FM/AFM heterobilayer films[18–24]. However, unlike the heterobilayer system, we observed a systematic enhancement in the coercive field with the cooling fields. A similar field-cooled exchange bias effect can also be observed in device D5 (S.I. Fig. S4). As exchange bias cannot arise solely from a simple FM phase without other exchange interactions, the exchange bias in the uncompensated MBT is related to a modification of the exchange coupling between the uncompensated FM layer and the compensated bulk AFM layer through the field cooling process.

To gain further insight into the field cooling effect, we parameterize the exchange bias field ($H_{EB} = (H_C^+ - H_C^-)/2$) and the total coercive field ($H_C = H_C^+ + H_C^-$) for the different cooling fields ($H_{FC}$), as summarized in Fig. 2e. We observed a sizeable $H_{EB}$ of up to -90 mT, along with a ~30% enhancement in $H_C$ for field cooling at −6 T. Furthermore, as shown in the inset of Fig. 2e, we found that the enhancement in $H_C$ is proportional to the $H_{EB}$. This indicates that the larger cooling fields lead to an improvement of the pinning effect in the bulk AFM layer[37], resulting in an enhancement of the $J_1/J$ ratio. Additionally, it is observed that the $H_{EB}$ and $H_C$ under positive cooling fields are smaller compared to those under negative cooling fields. This non-antisymmetric behavior under opposite directions of cooling fields may suggest the presence of a net moment in the inner bulk MBT, which originates from the Mn/Bi antisite defects[38]. The existence of this non-zero magnetization in the AFM bulk can enhance $H_C^+$ (reduce $H_C^-$) when they are in the same (opposite) direction with the applied magnetic field. Consequently, this contributes to the non-antisymmetric nature of the exchange bias under opposite magnetic fields. This phenomenon is reminiscent of the case observed in the oxidized-FGT/FGT/CrSe heterostructure[39], where the presence of net moment in the non-colinear CrSe layer is attributed to the non-antisymmetric exchange bias under cooling fields.

## Field training exchange bias

The effect of the exchange coupling strength between the compensated and uncompensated layers can be observed more directly through the exchange bias induced by field training. To demonstrate this effect, we compare the field training exchange bias in five 7SL MBT devices labeled as D2–D6 (device details can be found in S.I. Table S1). The field training experiments were conducted in the following sequence, as illustrated in Fig. 3a. The field sweeping was initiated at zero magnetic field and then set to a training field ($H_{FT}$), followed by a forward and backward field sweep. The next training field was then set, and a field sweep was performed without undergoing another cooling cycle. The temperature was kept constant at 2 K during the measurements. The cooling cycle was only applied when reversing the field train direction. This cooling cycle involved warming the sample to a temperature above the $T_N$ (typically to $T = 40$ K) to erase the memory of the Mn spins, followed by cooling to 2 K at zero magnetic field.

The $R_{yx}$ hysteresis loops under different field training conditions for the devices D2–D6 are shown in Fig. 3b–f. For device D2 (Fig. 3b), we observed a minimal exchange bias at a training field of −2 T, with slight asymmetry between negative and positive coercive fields. However, when a higher training field was applied, the symmetry

coercive field was preserved. It is worth noting that this device was prepared by a lithography-free bottom contact method in a dry box, which is expected to preserve the fresh outermost layer. Device D3 (Fig. 3c) exhibits a sizable exchange bias even at a small training field of 1 T. Increasing the training field to an intermediate strength further enhances the $H_{EB}$, as evidenced by the substantial extension of the negative coercive field to -0.70 T during positive field training at +2.5 T, while the positive coercive field remaining unaltered at -0.05 T. A similar phenomenon was observed when trained with a −2.5 T field, where the negative and positive coercive fields inverted. By expanding the field sweep range beyond ±4 T, the exchange bias could be reset, resulting in a symmetric and wider anomalous Hall loop centered around zero magnetic field. Device D4 (Fig. 3d) exhibits a similar field training exchange bias effect.

The field training protocol results in a more complex exchange bias feature in device D5. As shown in Fig. 3e, a low training field (<+2 T) does not induce a measurable exchange bias. However, the field training-induced exchange bias effect becomes evident in the intermediate training field range of +3 to +6 T. Surprisingly, both the exchange bias and total coercivity are amplified and reach their maximum values in this intermediate field training region. This behavior is consistent when the opposite training field is applied, with the exchange bias field showing an inverted sign, further indicating the negative exchange bias effect, as also suggested by the field cooling results (S.I. Fig. S4c). We noted that the field-cooled and field-trained exchange bias in D5 behave differently due to the different spin pinning mechanisms, where the mechanism for the field-training exchange bias will be discussed later. Similarly, a higher training field of +9 T can reset exchange bias and produce a symmetric anomalous Hall loop with a much smaller coercive field ($H_C$ ~ 0.2 T) in device D5. Finally, in device D6 (Fig. 3f), we observed a much smaller coercive field $H_C$ ~ 0.1 T with no exchange bias throughout the entire field training range.

To better evaluate the field training-induced exchange bias effect in the uncompensated MBT, we extract the $H_{EB}$ and $H_C$ as a function of the training field ($H_{FT}$) for the studied devices, as plotted in Fig. 3g–k. Despite variations in $H_{EB}$ and $H_C$ from device to device, we observed several consistent features in their field-trained exchange bias. Consistent with the field-cooled exchange bias, the field training results in a negative exchange bias across the entire field range. Interestingly, the $H_{EB}$ is maximized at an intermediate training field, while the exchange bias effect vanishes when the field sweep range is further expanded at a higher training field. We can therefore define the training field required to reset the exchange bias as the critical exchange bias field ($H_{EB}^c$).

## Simulations and the proposed mechanism

To gain insights into the mechanisms of the field-trained exchange bias effect, we performed micromagnetic simulations for the uncompensated MBT under different exchange coupling conditions. In Fig. 4, we compare the experimental and simulated field hysteresis loops without and with types I and II exchange bias. We first examine the field training response for the case of equivalent exchange coupling ($J_1 = J_i$). Figure 4b, c shows the simulated hysteresis loop in magnetization and the corresponding spin alignment, respectively, for the equivalent exchange coupling. As expected, the field training process does not result in any asymmetry or exchange bias in the hysteresis loop. However, as we reduce the exchange coupling of the first layer ($J_1$), we begin to observe the field-trained exchange bias effect. In the following, we discuss the origin of these two types of exchange bias.

The type I exchange bias occurs when the training field is within a range smaller than the $H_{EB}^c$, as demonstrated by device D3 (Fig. 4d). In our simulation results shown in Fig. 4e, we start from a strong external magnetic field of −9 T, which drives the MBT to a spin-aligned state where all the spins point down. As we sweep the field to −1 T, the MBT evolves to a ground minimization state with an antiparallel alignment of neighboring spins. Assuming weak coupling due to the surface

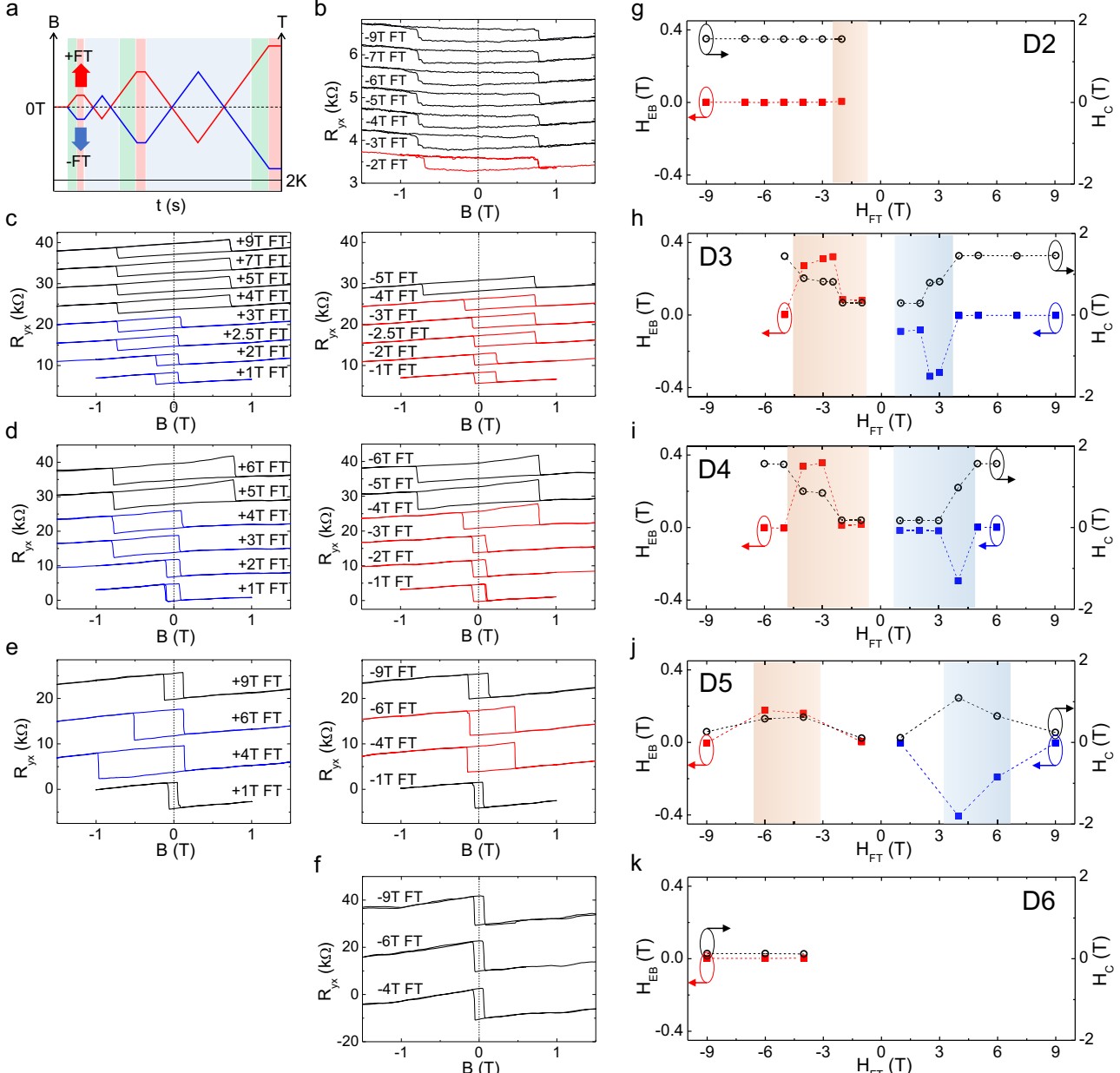

**Fig. 3 | Field training exchange bias. a** Illustration of the field training protocol for the setting of temperature and magnetic field as a function of time. Black, red, and blue curves represent the temperature, and the field sweep sequence for positive and negative field training, respectively. The green, red, and blue shades highlight the fieldset, field train, and field sweep regions, respectively. Plots of $R_{yx}$ hysteresis loops in the magnetic field under zero field initialization at positive ($+H_{FT}$) and negative ($-H_{FT}$) field training protocols for the different MBT devices of **b** D2, **c** D3, **d** D4, **e** D5, and **f** D6. All data were taken at 2 K. The $R_{yx}$ versus magnetic field curves in all panels are shifted vertically for comparison. The training fields for each curve are labeled in the figures. In **c**–**e**, the $R_{yx}$ curves for positive and negative field training are plotted separately in the left and right panels for clarity. The $R_{yx}$ hysteresis loops with exchange bias induced by the positive and negative field training are plotted in blue and red curves, respectively. Plots of the exchange bias field ($H_{EB}$) and the coercive field ($H_C$) as a function of the applied training field ($H_{FT}$) extracted from the $R_{yx}$ hysteresis loops of the respective 7SL MBT devices D2–D6 in (**g**–**k**), respectively. The shade areas in the $H_{EB}$ versus $H_{FT}$ plots highlight the training field regimes with the exchange bias effect.

degradation, we set the out-of-plane exchange coupling $J_1$ to be much smaller than $J_2$–$J_6$ (specifically, $J_1 = 0.05J_2$) in our simulation. As a result, the spin in the top layer prefers to flip itself, while the remaining six layers maintain their spin alignment due to the lower energy barrier when the magnetic field changes the sign to +1 T. When reversing the field sweep direction, the AFM exchange coupling between the first and second layers makes it easier for the top layer to flip back to the spin-down state, as illustrated in Fig. 4f. This led to the observed exchange bias. Under these conditions, the 7SL system behaves as a quasi-FM/AFM bilayer, with the top uncompensated layer acting as the

FM layer while the remaining six layers act as the AFM pinning layer. Applying a large magnetic field is similar to the field cooling process, driving the quasi-FM/AFM into a single domain state. However, such isothermal exchange bias is rare in conventional AFMs. What makes the uncompensated MBT unique is that when the positive field continues to increase to +9 T, it enters spin-flip states again, resetting the magnetization states to point upwards. As a result, a full hysteresis loop with a maximum field larger than the spin-flip field ($H_2$) exhibits no exchange bias, as shown in the top blue curve in Fig. 4e and the top panel in Fig. 4f, which is consistent with the simulation results.

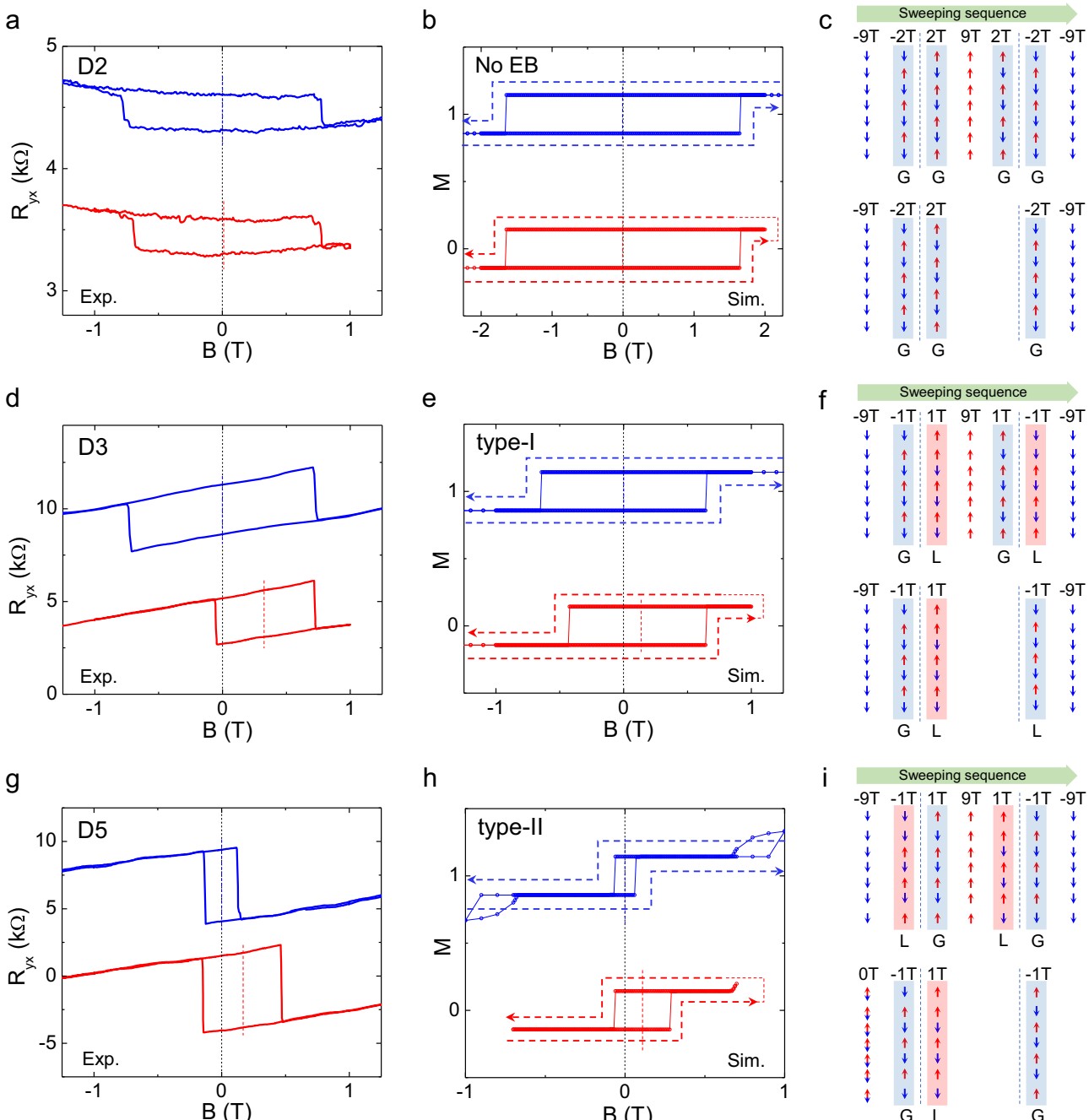

**Fig. 4 | Simulations and illustration of spin states for field training.** Comparison between the $R_{yx}$ hysteresis loop for the MBT devices **a** D2, **d** D3, and **g** D5, and the micromagnetic simulated magnetization loop under the different interlayer exchange coupling conditions of **b** no exchange bias ($J_1 = J_i$), **e** type I ($J_1 = 0.05J_2$), and **h** type II ($J_1 = 0.05J_2$, $J_6 = 0.7J_1$) exchange bias. The blue and red vertical dashed lines are drawn at the center of the hysteresis loop to indicate the horizontal shift of the coercive field due to exchange bias. Proposed field training mechanisms: schematic of the spin configurations in each layer under different magnetic fields to illustrate the (**c**) symmetric hysteresis loop, (**f**) type I, and (**i**) type II exchange bias effect. The top and bottom panels in (**c**), (**f**), and (**i**) display the spin states extracted from the top and bottom magnetization curves in (**b**), (**e**), and (**h**), respectively. The magnetic field labels at the top of each panel in (**c**), (**f**), and (**i**) indicate the sweeping sequence from left to right. The blue and red shades in (**c**), (**f**), and (**i**) indicate the global and local minimal states, respectively, in the AFM coupling of the uncompensated MBT.

The type II exchange bias, exemplified by device D5 (Fig. 4g), shares similarities with the type I exchange bias. However, unlike the type I case, where the full hysteresis loop deactivates the exchange bias by recovering the $H_C^-$ for the negative field training, the magnetization reset in the type II exchange bias is achieved by reducing the $H_C^+$ for the negative field training. This leads to a full loop (blue curve in Fig. 4g) with a coercivity smaller than the intermediate loop (red curve in Fig. 4g), which is highly nontrivial. To explain this observation, we

further assume that both the top and bottom surface exchange coupling are weakened due to the multidomain phase. This assumption is also supported by its smaller full field coercivity, which is consistent with our simulations in Fig. 1b. In our simulations with $J_1 = 0.05J_2$, and $J_6 = 0.7J_1$, the hysteresis loop under the same field training sequence is shown in Fig. 4h. Under these conditions when the field is swept from −9 T to −1 T, the bottom layer will preferentially flip first and induce the flipping of the 2nd to 5th layers to maintain antiparallel coupling. Thus, for the full

loop (−9 T to 9 T), the exchange field between the first and second layers assists the flip of the top layer for both the negative-to-positive and positive-to-negative field sweeps. However, for negative field training below the $H_{EB}^c$, as shown in the red curve in Fig. 4h with the spin structure of all the layers depicted in the bottom panel of Fig. 4i, the exchange field only assists the flip of the top layer for the positive-to-negative field sweep but hinders the flip for the negative-to-positive field sweep. This gives rise to the exchange bias with a larger coercive field for the intermediate training field compared to the symmetrical hysteresis loop at a smaller coercivity field for the full loop sweep.

The simulation results accurately capture the observed exchange bias features in our experiments. This suggests that the degree of weakening of the $J_S$ in the outermost layers can give rise to different types of field training exchange bias effects. Based on these simulations and experimental observations, we attribute the observed type I and type II exchange bias effects to different levels of oxidation, which result in a weakening of the exchange coupling strength between the outermost and the inner layers. Specifically, the type I exchange bias emerges when only the exchange coupling in the top outermost layer is weakened, while the type II exchange bias forms when both the top and bottom outermost layers are weakened, creating a quasi-FM/AFM sandwich structure. To further verify our interpretation, we performed additional experiments on device D4 nearly one month after the first cooling (S.I. Fig. S8). The results show a change in exchange bias behavior from type I to a type II-like effect. This agrees with our interpretation, where the type II exchange bias emerges in samples with more pronounced oxidation in the outermost layers. However, it is important to note that as $J_{1,6} \ll J_i$ is significantly weakened, oxidation can eventually affect the inner exchange coupling strengths ($J_2$–$J_5$). This can result in a gradient of exchange coupling as a function of depth. As $J_2$–$J_5$ gradually reduces, the quasi-FM/AFM structure is no longer sustained, leading to the vanishing of the exchange bias due to the lack of spin pinning from the bulk AFM. This explains the absence of exchange bias in D6, which exhibits a much smaller coercive field.

We further discuss the exchange bias effect in the zero-field initialization case in the field training protocol, which differs from the field cooling procedure in FM/AFM hetero-bilayers where the exchange bias is absent with zero-field cooling. Our SSM analysis (Fig. 2a) indicates that zero-field cooling can induce multidomain states in the uncompensated layer. To understand the effect of the

domain structure on the emerging exchange bias at zero field, we performed simulations of magnetization loops by varying the domain size while keeping the condition of the first layer exchange coupling $J_1 = 0.05J_2$ unchanged, as shown in Fig. 5. In Fig. 5b, the simulation result for the larger domain size (larger than the total simulation area) demonstrates that even when starting from 0 T with a maximum negative training field of −1 T, the net moment resulting from the uncompensated AFM coupling under a small field can still lead to a smaller but non-vanishing exchange bias. However, as the domain size decreases, the exchange bias under zero field initialization gradually diminishes. This is primarily due to the weakening of the effective AFM coupling strength caused by the multiple-domain structure. Consequently, the pinning effect from the AFM bulk on the surface uncompensated layer is reduced. In Fig. 5b, we observe a nearly symmetric hysteresis loop for the case of a small domain size. These simulation results are highly consistent with the experimental observations as shown in Fig. 5a, and also align with our observations of field training exchange bias effects, as discussed in Fig. 4f, i.

Based on the simulation results, we can then summarize the requirements for the exchange bias in the uncompensated antiferromagnet MBT as follows: (1) Difference in exchange coupling strength. There needs to be a disparity in the exchange coupling strength between the uncompensated layer and the compensated bulk to form the quasi-FM/AFM configuration. (2) Presence of single or large domain structure. The presence of a multidomain structure weakens the effective magnetic moment in the inner AFM bulk layer, which can deteriorate the exchange bias under zero or low-field training. These requirements are in contrast to standard FM/AFM hetero-bilayers[18–24], where the exchange bias effect typically relies on a strong interface exchange coupling. Based on these findings, we can infer a good correlation between the critical training field and the exchange coupling in the uncompensated layer. This suggests that continuous tuning of the field-trained exchange bias is possible in the uncompensated AFM. Additionally, the observed persistence of the field training exchange coupling over multiple cycles and a wide range of magnetic fields allows a systematic control of the interlayer exchange coupling. The control of exchange coupling strength between the outermost and inner layers can be achieved through various methods, such as oxidation[40], intercalation[28,41], strain[42], hydrostatic pressure[22], and other techniques. One straightforward method to control the exchange

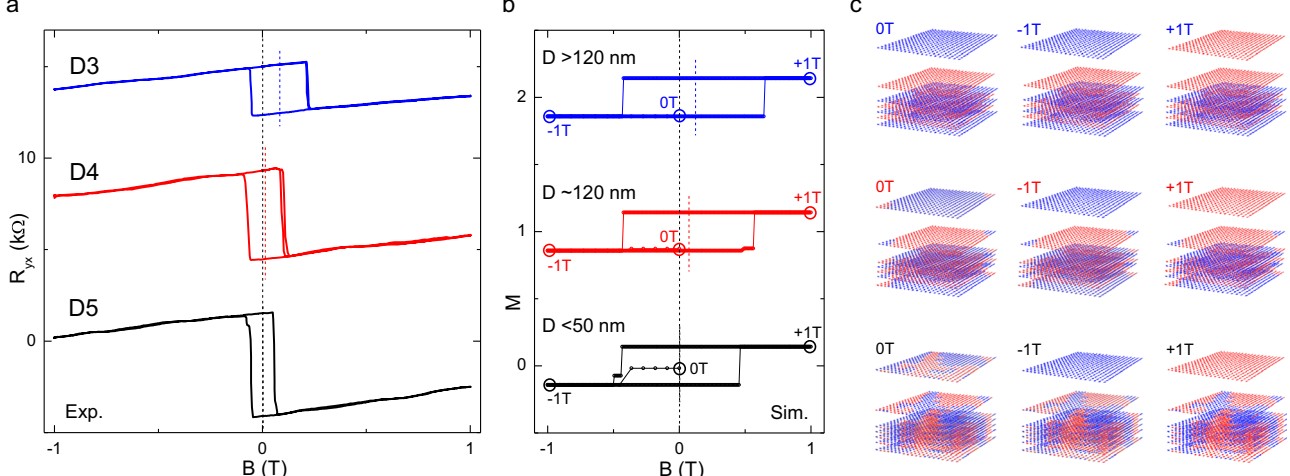

**Fig. 5 | Illustration of domain structures for zero field initialization.** Comparison between **a** $R_{yx}$ hysteresis loop for the 7SL MBT devices D3–D5 and **b** micromagnetic simulated magnetization loops with different domain states at 0 T field for field training protocol under the zero-field initialization and followed by a field sweep at ±1 T. The vertical dashed lines are drawn at the center of the hysteresis loop to indicate the horizontal shift of the coercive field due to exchange bias. **c** Simulated spin states in each layer at the 0 T initialization, field sweep at −1 T and +1 T for the different domain sizes, D, corresponding to the magnetization loops in (**b**). The spin-up and spin-down states illustrated in (**c**) are presented by the red and blue arrows.

coupling strength is by systematically controlling the exposure to an oxygen agent in the oxidation process[40]. Intercalation, on the other hand, can be utilized to adjust the interlayer coupling strength by tuning the interlayer distance through the insertion of an intercalated layer. For instance, the intercalation of alkali ions by filling the van der Waals gap has been found to effectively modulate the $J_s$[43]. Similarly, uniaxial tensile strain and hydrostatic pressure can enhance $J_s$ by mechanically reducing the interlayer distance in van der Waals layers, as observed in 2D van der Waals magnets.

## Summary

In summary, our study presents a comprehensive investigation of the exchange bias effect in uncompensated antiferromagnetic MBT devices. The presence of both field cooling and field training-induced exchange bias in the single-odd SL MBT indicates the existence of exchange interaction between the uncompensated and compensated AFM layers, leading to the formation of a quasi-FM/AFM structure. By manipulating the exchange coupling strength in the uncompensated FM layer, we observed the emergence of a local minimum state, as confirmed by both micromagnetic simulations and experimental results. This provides a wide range of tunability for exchange bias and coercive fields in the odd SL MBT. The unconventional field training protocol results in a marginally large exchange bias field, which is induced by the intimate exchange coupling within the single-layer system. Furthermore, this exchange bias effect remains robust over multiple field sweeps, making it highly reliable for spintronic-related applications.

## Methods

### Scanning SQUID measurement

The measurements of magnetization in the uncompensated layer of MBT were performed by scanning SQUID (superconducting quantum interference device) microscopy (SSM). The SQUID, consisting of a loop of superconducting Josephson junctions, exhibits high sensitivity to magnetic flux, allowing us to image weak local magnetic fields emanating from the MBT flake. Our SQUID sensor comprises two Nb pickup loops and field coil pairs arranged in a gradiometric structure. The gradiometric design effectively eliminates the influence of the background magnetic field, ensuring that the pickup loop exclusively measures the magnetic response originating from the sample. The pickup loop and field coil are encased in Nb shielding, effectively directing the flux exclusively through the loop while preventing it from passing through the gaps between the leads. The pickup loop had an inner radius of 0.4 μm, enabling submicron spatial resolution. The scan height is ~260 nm. During the scanning process of the SQUID sensor over the sample surface, we capture and record the magnetic flux through the pickup loop. The local dc magnetic response is recorded in magnetometry mode with unit $\Phi_0$. Here $\Phi_0$ corresponds to the quantum magnetic flux h/2e, where h is the Planck constant and e is the elementary charge.

### Devices and transport measurement

MBT samples were prepared by mechanical exfoliation from the parent MBT crystals[44]. The MBT flakes were exfoliated in an argon-filled glovebox. The thickness of the MBT flakes was checked by a Bruker dimension atomic force microscope and compared with the optical contrast. The MBT devices are fabricated into a Hall-bar geometry with Si/SiO$_2$ gate. All devices, except device D2, were fabricated using the standard electron beam lithography, followed by Cr/Au deposition for contact electrodes. During the transport for electron beam lithography and metal deposition processes, the MBT flakes are protected by PMMA. Device D2 was fabricated by transferring the MBT flake onto a pre-patterned Hall bar electrode using polypropylene carbonate (PPC). The magnetotransport measurements were carried out in a variable temperature physical property measurement system (PPMS) at a base temperature of 2 K. The longitudinal ($R_{xx}$) and Hall ($R_{yx}$) resistance

signals were recorded by SR830 lock-in amplifiers, and the gate voltage was controlled by a Keithley K2400 voltage-source meter. Unless otherwise specified, all the data were collected at the temperature of 2 K.

### Micromagnetic simulations

Micromagnetic simulations were performed using the standard OOMMF[45] extensible solver (OXS). To achieve the construction of A-type AFM MBT, we set the in-plane ferromagnetic intralayer coupling with positive exchange stiffness $A_{in}$ and the out-of-plane antiferromagnetic interlayer coupling with negative exchange stiffness $A_{out}$. The system size in the simulation is fixed at $100 \times 100 \times 7$ nm$^3$ for a total of 7 layers with a mesh size $5 \times 5 \times 1$ nm$^3$. The material parameters in the simulated block for type I exchange bias (device D3) are set with in-plane exchange stiffness $A_{in} = 2.6 \times 10^{-13}$ J/m, $A_{out} = -1.5 \times 10^{-13}$ for the couplings between layers 2 and 6 and $A_{out} = -7.5 \times 10^{-14}$ for coupling between layer 1 (top layer) and layer 2, the saturation magnetization ($M_s = 1.4 \times 10^5$ A/m) and the perpendicular anisotropy $K_u = 5 \times 10^4$ J/m$^3$. The material parameters in the simulated block for type II exchange bias (device D5) are set with in-plane exchange stiffness $A_{in} = 4.5 \times 10^{-14}$ J/m, $A_{out} = -1.5 \times 10^{-13}$ for the couplings between layers 2 and 5, $A_{out} = -7.5 \times 10^{-14}$ for coupling between layer 1 (top layer) and layer 2, and $A_{out} = -1.05 \times 10^{-13}$ for the coupling between layer 6 and layer 7 (bottom layer), the saturation magnetization ($M_s = 1.4 \times 10^5$ A/m) and the perpendicular anisotropy $K_u = 2.8 \times 10^4$ J/m$^3$.

## Data availability

The data supporting the findings of this study are available from the corresponding authors upon request.

## Code availability

The codes used for the numerical simulation are available from the corresponding authors upon request.

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

## Acknowledgements

This work was supported by the National Science Foundation, the Quantum Leap Big Idea under Grant No. 1936383, and the U.S. Army Research Office MURI program under Grant No. W911NF-20-2-0166 and No. W911NF-16-1-0472. Support for crystal growth and characterization was provided by the National Science Foundation through the Penn State 2D Crystal Consortium-Materials Innovation Platform (2DCC-MIP) under NSF cooperative agreement DMR-2039351. Z.Q.M. also acknowledges the support from NSF under Grant No. 2211327. H.M. and K.A.M. were supported by the US Department of Energy, Office of Basic Energy Sciences, Division of Materials Sciences and Engineering under award DE-AC02-76SF00515. The SSM experiments utilized equipment in the Stanford Nano Shared Facilities, funded by the National Science Foundation under the award ECCS-2026822. S.K.C. acknowledges the support from Beijing Natural Science Foundation under Grant No. IS23022.

## Author contributions

S.K.C., Y.C. and K.L.W. planned the experimental project. S.H.L., Y.W., and Z.M. prepared the bulk crystals. S.K.C., M.T. and T.H.Y. fabricated the devices and conducted the transport measurements. H.M. and K.A.M. performed the SSM measurements and analyzed the data. Y.C. and B.D. performed the theoretical simulations, analysis, and modeling. S.K.C., Y.C., H.M. and K.L.W. wrote the paper. All authors discussed the results and commented on the paper.

## Competing interests

The authors declare no competing interests.
