## [Peer Review File · Nature Communications]

Reviewers' Comments:

Reviewer #1:

Remarks to the Author:

I have carefully read the manuscript by Su Kong Chong et al. which studies exchange bias in uncompensated antiferromagnetic MnBi₂Te₄ devices. This is an extremely interesting paper. Notably, the authors report the presence of field cooling and field training-induced exchange bias in single-odd septuple-layer MnBi₂Te₄ and attributes the quasi-FM/AFM structure to the formation of exchange interactions between the uncompensated and compensated AFM layers through simulations and experiments result. I found the work interesting, but there are several key issues which must be addressed before any decision can be made.

1. In view of the content studied in the article, the number of material layers has a great influence on the formation of quasi-FM/AFM structures. The author mentioned that the 6 devices are all 7 septuple-layer devices, so how to obtain the material thickness accurately? Please provide relevant characterization data.
2. The author proposed that the exchange bias effect can be modulated by the coupling strength between the uncompensated FM layer and the compensated bulk AFM layer and domain structure, so do the authors provide some methods to quantitatively modulate the coupling strength and domain structure?
3. Both the top layer and the bottom layer belong to the outermost layer. How does the author judge which layer can be used as the outermost layer to couple with compensated AFM layers to form a quasi-FM/AFM structure. In addition, due to the existence of the bottommost layer and the topmost layer, whether a quasi-FM/AFM/quasi-FM three-layer structure will be formed will affect the overall results.
4. Numbers (i), (ii), (iii), (iv), and (v) used for the subfigures of Figure 2a. However, the authors use numbers (i), (ii), and (iii) to describe this part of the results in line 110, which leads to misunderstanding and needs to be improved by the author.
5. Fig. 2f is mentioned in line 127, but this part is not covered in Figure 2.
6. For panels c and d in Figure 1, please provide more detailed labels for the x-axis and y-axis.

Reviewer #2:

Remarks to the Author:

This work studies the exchange bias effect in the uncompensated (odd-layer) antiferromagnetic (AFM) topological material MnBi₂Te₄. The odd-SL MnBi₂Te₄ is expected to behave as a quasi-FM/AFM bilayer, which can lead to the exchange interaction depending on the strengths of the interlayer coupling. Two kinds of methods are utilized to measure the exchange bias anomalous Hall effect, i.e. field cooling and field training, and an exchange bias of up to ~400 mT is observed. The exchange bias effect in the 7-SLs MnBi₂Te₄ thin film is very complex, and different from the traditional FM/AFM bilayers. The authors proposed comprehensible mechanisms to explain the most of the experimental phenomena. Before considering publication of the manuscript, there are several issues should be addressed.

1. My first concern is the sample dependence of the exchange bias in the 7-SLs MnBi₂Te₄. The six MnBi₂Te₄ devices measured in this work show the distinct behaviors (listed in Table S1), which are fabricated with different methods. According to the different interlayer exchange coupling, there will be three kinds of situations, i.e. no exchange bias, type I and type II exchange bias. It seems that the behavior of exchange bias is independent of the fabrication method. What cause the devices to exhibit the different exchange bias behavior?
2. In Fig. 1, the authors confirm the quasi-FM/AFM bilayer structure in 7-SLs MnBi₂Te₄ in the case of $J_1 \ll J$, and the weak J_1 would lead to a reduction in the coercive field. In device D2 (Fig. 3b), the coercive field is large, and the exchange bias diminishes. This behavior is consistent with the above mechanism. In device D6 (Fig. 3f), the coercive field is smaller than other devices, but the exchange bias effect disappears throughout the entire field training range. It seems to contradict the above mechanism. Please discuss this clearly.
3. The multidomain structure, which weakens the effective magnetic moment in the uncompensated layer, is necessary for the exchange bias effect in the uncompensated MnBi₂Te₄. But in Fig. 5, the authors argue that the small domain size would lead to the diminishment of the

exchange bias under zero field initialization, due to the weakening of the effective AFM coupling strength caused by the multiple domain structure. I'm confused about the effect of the multidomain structure.

4 The title of this work is 'Intrinsic exchange biased anomalous Hall effect in an uncompensated antiferromagnet MnBi₂Te₄'. What is the meaning of the 'intrinsic exchange bias'? It is not addressed in the main text.

5. In Figs. 1 (d, e), under the cooling fields with the same value but opposite directions, the magnitude of exchange bias field is different. In the previous works, it is usually the same. Please discuss the possible reasons.

6. In device D5, there is a large difference in the exchange bias field under different measurement conditions (field cooling and field training). What causes such a large difference?

Reviewer #3:

This work reports a material with tunable exchange bias (EB). Typically, EB is found in AFM/FM heterostructure, where the AFM layer is used to pin the FM layer. Here, the authors show clear EB in an uncompensated AFM flake. Transport measurements demonstrate that a field training protocol can tune the EB in situ. The authors propose a model to explain their model. This model is based on some form of sample degradation that alters the exchange coupling between the top and neighboring layers. Two types of EB are observed; one is related to the presence of magnetic domains observed by scanning SQUID microscopy. I believe the work is novel and exciting, however, I think the paper could be clearer and better presented. Therefore, I recommend publishing this work in Nature Communications after these minor revisions.

1. I understand that explaining this type of data is challenging, but I believe the authors could be easier to understand. For example, the training protocol illustrated in Figure 3a is not clear. For the +FT, what is the negative field reached? Is it the same in absolute value, or is the field sweep asymmetric? The protocol is a rather important point and should be straightforward to understand. I recommend reviewing the entire article and thinking of a simpler way to present the data.

2. Many figures are also not clear. For example, in Figure 3h, it is unclear what data correspond to the different y-axis. In Figures 4c,f, and i, each arrow represents the magnetization of each layer but it is unclear what each row and column represents. Are they part of a sequence in time? Also, the resolution of some panels is too low.

3. Can the authors explain what causes some samples to behave like type I and some like type II? Did the author consider capping the flake to exclude oxidation effects?

4. I believe more relevant previous work should be cited.

4.1 On lines 59-60, the authors mention that no EB was found in thin films of Co₃Sn₂S₂. The authors should cite the work where the EB absence was discovered (see Noah *et al.* Phys. Rev. B 105, 144423 (2022), <https://link.aps.org/doi/10.1103/PhysRevB.105.144423>). This work also tunes the EB with similar training protocols and mentions the presence of uncompensated moments at magnetic domain walls as a potential underlying physical mechanism.

4.2 I consider mentioning Nayak *et al.* Nat. Mater. 14,679 (2015), where tunable EB was also reported in a single compound, not a heterostructure.

4.3 Finally, I would also consider Wang *et al.* PRL 106, 077203, <https://link.aps.org/doi/10.1103/PhysRevLett.106.077203>)

Responses to Reviewers' comments:

Reviewer #1 (Remarks to the Author):

I have carefully read the manuscript by Su Kong Chong et al. which studies exchange bias in uncompensated antiferromagnetic MnBi₂Te₄ devices. This is an extremely interesting paper. Notably, the authors report the presence of field cooling and field training-induced exchange bias in single-odd septuple-layer MnBi₂Te₄ and attributes the quasi-FM/AFM structure to the formation of exchange interactions between the uncompensated and compensated AFM layers through simulations and experiments result. I found the work interesting, but there are several key issues which must be addressed before any decision can be made.

Response: We thank the reviewer for appreciating this work and for the important comments and suggestions to help us improve this manuscript.

1. In view of the content studied in the article, the number of material layers has a great influence on the formation of quasi-FM/AFM structures. The author mentioned that the 6 devices are all 7 septuple-layer devices, so how to obtain the material thickness accurately? Please provide relevant characterization data.

Response: We have included the characterization data for all the devices utilized in this study. The thickness of these flakes was estimated based on their optical contrast (before device fabrication) and subsequently confirmed using atomic force microscopy (after device fabrication). The optical images and atomic force microscopy data of these samples can be found in S.I. Fig. S1. Moreover, the thickness of these devices was further validated through magnetic transitions. It is worth noting that the magnetic transition fields exhibit a strong dependence on the thickness, allowing for clear differentiation between even and odd layers, as demonstrated in the magnetic circular dichroism data in our previous publication (Phys. Rev. X 11, 011003 (2021)). Thus, by comparing the magnetic transition fields for spin-flop and spin-flip transitions, we can further verify the thicknesses, as illustrated in S.I. Fig. S3.

2. The author proposed that the exchange bias effect can be modulated by the coupling strength between the uncompensated FM layer and the compensated bulk AFM layer and domain structure, so do the authors provide some methods to quantitatively modulate the coupling strength and domain structure?

Response: Our experimental results, as well as theoretical models, indicate that the coupling strength plays a significant role in the exchange bias behavior of MBT uncompensated AFMs. In response to the reviewer's suggestions, we have included a discussion on possible methods to modulate the coupling strength in the uncompensated layer of MBT. One potential approach to achieve control is by manipulating the oxygen exposure in a controlled environment. By carefully adjusting the relevant parameters, such as the O₂ partial pressure, duration, etc., it may be possible to systematically introduce oxidation to reduce the coupling strength. Additionally, ion intercalation and strain/pressure have been shown to be effective in tuning the exchange coupling strength in 2D magnets (Adv. Phys. Res. 2, 2200106 (2023)). We have incorporated the discussion in the revised manuscript.

3. Both the top layer and the bottom layer belong to the outermost layer. How does the author judge which layer can be used as the outermost layer to couple with compensated AFM layers to form a quasi-FM/AFM structure. In addition, due to the existence of the bottommost layer and the topmost layer, whether a quasi-FM/AFM/quasi-FM three-layer structure will be formed will affect the overall results.

Response: We fully acknowledge the reviewer's observation that both the top and bottom layers belong to the outermost layer and can contribute to the exchange bias in this quasi-FM/AFM structure. Our simulation results support this notion, demonstrating that the weakening of the exchange coupling in both the bottommost and topmost layers can give rise to a distinct exchange bias effect in odd-SL MBT, which we have referred to as type II exchange bias. This finding is explicitly discussed in the main text. Specifically, in this scenario, the exchange coupling of J_1 (top layer) and J_6 (bottom layer) is simultaneously weakened.

4. Numbers (i), (ii), (iii), (iv), and (v) used for the subfigures of Figure 2a. However, the authors use numbers (i), (ii), and (iii) to describe this part of the results in line 110, which leads to misunderstanding and needs to be improved by the author.

Response: We appreciate the reviewer's feedback regarding the inconsistency between the figure numbering and the text. To address this concern and prevent any confusion, we have made revisions to the paragraph structure in the revised manuscript. Specifically, we have removed the numbers (i), (ii), and (iii) to create a clearer and more cohesive presentation.

5. Fig. 2f is mentioned in line 127, but this part is not covered in Figure 2.

Response: We have identified the error in our initial manuscript regarding the citation of the Figure 2. The correct figure to reference is S.I. Fig. S3, not the figure that was originally cited. We have made the necessary correction in the revised manuscript. We would like to thank the reviewer for bringing this error to our attention.

6. For panels c and d in Figure 1, please provide more detailed labels for the x-axis and y-axis.

Response: We have made clarifications regarding the x-axis and y-axis in Fig. 1c and d. Specifically, the x-axis represents the spin angles in layer 1, while the y-axis represents the spin angles in layers 2-7 of the 7SL MBT. We have added detailed labels for the x-axis and y-axis in Fig. 1c and d, and have included a description in the figure caption to provide additional context.

Reviewer #2 (Remarks to the Author):

This work studies the exchange bias effect in the uncompensated (odd-layer) antiferromagnetic (AFM) topological material MnBi₂Te₄. The odd-SL MnBi₂Te₄ is expected to behave as a quasi-FM/AFM bilayer, which can lead to the exchange interaction depending on the strengths of the interlayer coupling. Two kinds of methods are utilized to measure the exchange bias anomalous Hall effect, i.e. field cooling and field training, and an exchange bias of up to ~400 mT is observed. The exchange bias effect in the 7-SLs MnBi₂Te₄ thin film is very complex, and different from the traditional FM/AFM bilayers. The authors proposed comprehensible mechanisms to explain the most of the experimental phenomena. Before considering publication of the manuscript, there are several issues should be addressed.

Response: We thank the reviewer for providing their critical comments and professional insights to help us improve the shortcoming in this manuscript.

1. My first concern is the sample dependence of the exchange bias in the 7-SLs MnBi₂Te₄. The six MnBi₂Te₄ devices measured in this work show the distinct behaviors (listed in Table S1), which are fabricated with different methods. According to the different interlayer exchange coupling, there will be three kinds of situations, i.e. no exchange bias, type I and type II exchange bias. It seems that the behavior of exchange bias is independent of the fabrication method. What cause the devices to exhibit the different exchange bias behavior?

Response: We concur with the reviewer's assessment that the behavior of exchange bias is independent of the fabrication method. However, we would like to clarify that the observed exchange bias in our 7SL MBT system is likely a result of the weakening of the exchange coupling strength in the outermost layer due to exposure to ambient conditions during the fabrication process. It is therefore reasonable to assume that the different levels of oxidation can lead varying degrees of weakening in the exchange coupling strength between the outermost and inner layers, giving rise to different types of exchange bias. Our simulation results, which consider the weakening of the exchange coupling strength in (i) the top outermost layer, J_1 , and (ii) both the top and bottom outermost layers, J_1 and J_6 , for a 7SL MBT, successfully capture the different types of exchange bias observed experimentally, consistent with type I and type II exchange bias behaviors. This supports our interpretation that the different types of exchange bias can be attributed to varying levels of oxidation and, consequently, different exchange coupling strengths, as indicated by our theoretical calculations. Moreover, as an additional confirmation, we performed measurements on device D4 approximately one month after the initial cooling, during which it was kept in glovebox with O₂ level ~1 ppm. These measurements revealed a change in the exchange bias from type I to a type II like, as depicted in S.I. Fig. S8 in the revised manuscript. This information further strengthens our interpretation, as it indicates that type II exchange bias emerges in samples with more pronounced oxidation in the outermost layers. We have incorporated a discussion on the reasons for the different types of exchange bias in the revised manuscript.

2. In Fig. 1, the authors confirm the quasi-FM/AFM bilayer structure in 7-SLs MnBi₂Te₄ in the case of $J_1 \ll J$, and the weak J_1 would lead to a reduction in the coercive field. In device D2 (Fig. 3b), the coercive field is large, and the exchange bias diminishes. This behavior is consistent with the above mechanism. In device D6 (Fig. 3f), the coercive field is smaller than

other devices, but the exchange bias effect disappears throughout the entire field training range. It seems to contradict the above mechanism. Please discuss this clearly.

Response: We apologize for the missing the explanation regarding the disappearance of exchange bias in D6. To clarify, the smallest coercive field in D6 suggests that this sample has a much weaker exchange coupling strength compared to the others, as indicated by our model (Figure 1b). It is important to note that as J_1 and J_6 are suppressed to much smaller values compared to J_i , the exchange coupling strength J_2 - J_5 are also affected. This results in a gradient of exchange coupling strength as a function of depth with the sample. As the difference in exchange coupling strength between the outermost and inner layers decreases, the quasi-FM/AFM structure becomes less stable. Consequently, the exchange bias vanishes. This can also be understood by examining the total energy, as illustrated in Fig. 1c and d. In the case of D6, the global minimum transitions to a local minimum, resulting in the disappearance of the pinning state and the exchange bias. We have incorporated this discussion in the revised manuscript for further clarity.

3. The multidomain structure, which weakens the effective magnetic moment in the uncompensated layer, is necessary for the exchange bias effect in the uncompensated MnBi₂Te₄. But in Fig. 5, the authors argue that the small domain size would lead to the diminishment of the exchange bias under zero field initialization, due to the weakening of the effective AFM coupling strength caused by the multiple domain structure. I'm confused about the effect of the multidomain structure.

Response: We apologize for any confusion caused by our previous statement regarding the effect of the multidomain structure. Upon reviewing our simulation results, we have clarified that reducing the domain size can indeed lead to the disappearance of the exchange bias under zero field initialization. This occurs due to the weakening of the effective AFM coupling strength caused by the presence of multiple domains within the sample. To avoid any further confusion, we have modified our statement on the multidomain structure. The revised statement now reads as follows: 'The presence of a multidomain structure weakens the effective magnetic moment in the inner AFM bulk layer, which can deteriorate the exchange bias under zero or low field training.' We apologize for any inconvenience caused by the previous statement and appreciate your understanding.

4 The title of this work is 'Intrinsic exchange biased anomalous Hall effect in an uncompensated antiferromagnet MnBi₂Te₄'. What is the meaning of the 'intrinsic exchange bias'? It is not addressed in the main text.

Response: We clarify that the intrinsic exchange bias here refers to the exchange bias effect that occurs within a single compound, rather than being influenced by other layers or substrate effects. Moreover, we acknowledge that the term 'intrinsic exchange bias' is not being introduced for the first time in our work. It has been previously used in the literature to describe the exchange bias that arises in a nominally single component ferro/ferri-magnetic material grown on a non-magnetic substrate, as discussed in the references J. Am. Chem. Soc. 145, 20041–20052 (2023) and Phys. Rev. B 103, 224405 (2021). We have added a definition and explanation of the term 'intrinsic exchange bias' to the introduction section of the revised manuscript.

5. In Figs. 1 (d, e), under the cooling fields with the same value but opposite directions, the magnitude of exchange bias field is different. In the previous works, it is usually the same. Please discuss the possible reasons.

Response: We appreciate the reviewer for bringing up the discrepancy between our field-cooled exchange bias results and previous studies. We would like to clarify that the exchange bias observed in the 7SL MBT is distinct from conventional FM/AFM systems. Unlike in conventional FM/AFM systems where only a horizontal shift in the hysteresis loop is observed under field cooling, the exchange bias in MBT is accompanied by an enhancement in the coercive field. We propose that the non-antisymmetric behavior observed in the field-cooled exchange bias may originate from a net moment within the inner AFM bulk of MBT. This phenomenon is quite common in MBT due to the presence of Mn/Bi antisite defects, which have been detected in MBT bulk crystals and can result in a sparse net magnetization. This non-zero magnetization within the AFM bulk can enhance H_C^+ in the same direction as the applied magnetic field (reducing H_C^- in the opposite direction). As a result, this contributes to the non-antisymmetric nature of the exchange bias under opposite magnetic field. This behavior is similar to the case of the oxidized-FGT/FGT/CrSe heterostructure, as reported in *Adv. Mater.* 34, 2105266 (2022), where presence of a net moment in the non-collinear CrSe layer is attributed to the non-antisymmetric exchange bias observed under opposite cooling fields. We would like to note that further investigation is needed to fully understand the detailed mechanism behind this non-antisymmetric exchange bias in the 7SL MBT system and to unravel the complexities of the field cooling mechanisms. We have included a discussion of this in the revised manuscript.

6. In device D5, there is a large difference in the exchange bias field under different measurement conditions (field cooling and field training). What causes such a large difference?

Response: We would like to clarify that the spin pinning mechanisms during field cooling and field training can be different, and therefore, they may not be equivalent to each other. In the case of field training, the exchange bias effect is induced by establishing a pinning AFM layer through the application of a setting field. As the training process involves a continuous sweeping of magnetic field without undergoing thermal cycling, the pinning due to the previous training field is unavoidable. On the other hand, during field cooling, the magnetic field is applied above the Neel temperature (T_N), allowing the FM-AFM interface to be pinned, thereby inducing the exchange bias effect. The spin pinning is therefore less affected by the previous cooling field. Based on our observations, we have noticed that the field training induced exchange bias is generally less sensitive to changes in the magnetic field until the next more stable pinning state is achieved. In contrast, the field cooling-induced exchange bias is more sensitive to the cooling fields, and the exchange bias field can be continuously adjusted by changing the cooling fields. Therefore, we speculate that the significant difference in the field cooling and field training exchange bias observed in device D5 is mainly due to the involvement of different pinning mechanisms. Furthermore, we acknowledge that a more systematic comparison between the field cooling and field training mechanisms is required before drawing more solid conclusions.

Reviewer #3 (Remarks to the Author):

This work reports a material with tunable exchange bias (EB). Typically, EB is found in AFM/FM heterostructure, where the AFM layer is used to pin the FM layer. Here, the authors show clear EB in an uncompensated AFM flake. Transport measurements demonstrate that a field training protocol can tune the EB in situ. The authors propose a model to explain their model. This model is based on some form of sample degradation that alters the exchange coupling between the top and neighboring layers. Two types of EB are observed; one is related to the presence of magnetic domains observed by scanning SQUID microscopy. I believe the work is novel and exciting, however, I think the paper could be clearer and better presented. Therefore, I recommend publishing this work in Nature Communications after these minor revisions.

Response: We thank the reviewer for appreciating this work and for the important comments and suggestions to help us improve this manuscript.

1. I understand that explaining this type of data is challenging, but I believe the authors could be easier to understand. For example, the training protocol illustrated in Figure 3a is not clear. For the +FT, what is the negative field reached? Is it the same in absolute value, or is the field sweep asymmetric? The protocol is a rather important point and should be straightforward to understand. I recommend reviewing the entire article and thinking of a simpler way to present the data.

Response: We appreciate the reviewer for bringing up the issue regarding the presentation of the field training protocol. We would like to clarify that the field training protocol follows a symmetry route with the same absolute value. For example, once the +FT state is reached, a symmetry field sweep is performed from +FT to -FT and then back to +FT. The subsequent field training is set by continuing to a higher +FT state. To address this concern, we have made modifications to the illustration in Figure 3a for a clearer explanation in the revised manuscript.

2. Many figures are also not clear. For example, in Figure 3h, it is unclear what data correspond to the different y-axis. In Figures 4c,f, and i, each arrow represents the

magnetization of each layer but it is unclear what each row and column represents. Are they part of a sequence in time? Also, the resolution of some panels is too low.

Response: We apologize for the unclear presentation in our previous version. We have made some improvements to address these issues. In Figures 3g-k, we have added indicators for the different y-axes to make it easier for readers to understand the data. Additionally, in Figures 4c, f and i, we have included indicators and additional explanations in the figure caption. Specifically, we would like to clarify that the different magnetic field labels at the top of each figure indicate the sweeping sequence in time. Furthermore, the top and bottom panels in Figures 4c, f and i display spin states in the sweeping sequence, which are extracted for the top and bottom simulated magnetization curves shown in Figures 4b, e and h, respectively. Lastly, to ensure the high resolution of each panel, we have uploaded the pdf version of the figures in the revised manuscript.

3. Can the authors explain what causes some samples to behave like type I and some like type II? Did the author consider capping the flake to exclude oxidation effects?

Response: We would like to clarify that the observed exchange bias in our 7SL MBT system is likely a result of the weakening of the exchange coupling strength in the outermost layer due to exposure to ambient conditions during the fabrication process. It is reasonable to assume that different levels of oxidation could lead to varying degrees of weakening in the exchange coupling strength between the outermost and the inner layers, resulting in different types of exchange bias. Our simulation results, which consider the weakening of exchange coupling strength in the top outermost layer (J_1) and both top and bottom outermost layers (J_1 and J_6) for a 7SL MBT, successfully capture the different types of exchange bias observed experimentally, consistent with type I and II behaviors. Therefore, we attribute the different types of exchange bias to the varying levels of oxidation and subsequent differences in exchange coupling strength, as indicated by our theoretical calculations. To further validate this interpretation, we performed measurements on device D4 approximately one month after the initial cooling, during which it was kept in glovebox with O_2 level ~ 1 ppm. These measurements revealed a change in the exchange bias from type I to a type II like, as depicted in S.I. Fig. S8 in the revised manuscript. This information further strengthens our interpretation, as it indicates that type II exchange bias emerges in samples with more pronounced oxidation in the outermost layers. Furthermore, when employing a proper capping and bottom contact, we observed a large hysteresis loop without any field training exchange bias, as illustrated in Fig. 3b in the main text. We have included a discussion on the reasons for the different types of exchange bias in the revised manuscript.

4. I believe more relevant previous work should be cited.

4.1 On lines 59-60, the authors mention that no EB was found in thin films of $Co_3Sn_2S_2$. The authors should cite the work where the EB absence was discovered (see Noah *et al.* Phys. Rev. B 105, 144423 (2022), <https://link.aps.org/doi/10.1103/PhysRevB.105.144423>). This work also tunes the EB with similar training protocols and mentions the presence of uncompensated moments at magnetic domain walls as a potential underlying physical mechanism.

Response: We apologize for missing this important reference. We have cited it in the revised manuscript.

4.2 I consider mentioning Nayak *et al.* Nat. Mater. 14,679 (2015), where tunable EB was also reported in a single compound, not a heterostructure.

Response: We have cited this paper in the revised manuscript.

4.3 Finally, I would also consider Wang et al. PRL 106, 077203, <https://link.aps.org/doi/10.1103/PhysRevLett.106.077203>)

Response: We have cited this paper in the revised manuscript.

Reviewers' Comments:

Reviewer #1:

Remarks to the Author:

The authors have entirely addressed my concerns with new data and clearer text and explanations. I applaud their efforts and strongly support publication of this highly interesting work in Nature Communications.

Reviewer #2:

Remarks to the Author:

The authors have addressed my comments properly.

Reviewer #3:

Remarks to the Author:

The authors properly answered all my comment. I now recommend it for publication in Nature Communications.